# Referential communication in heterogeneous communities of pre-trained visual deep networks

## Abstract

As large pre-trained image-processing neural networks are being embedded in autonomous agents such as self-driving cars or robots, the question arises of how such systems can communicate with each other about the surrounding world, despite their different architectures and training regimes. As a first step in this direction, we systematically explore the task of *referential communication* in a community of heterogeneous state-of-the-art pre-trained visual networks, showing that they can develop, in a self-supervised way, a shared protocol to refer to a target object among a set of candidates. This shared protocol can also be used, to some extent, to communicate about previously unseen object categories of different granularity. Moreover, a visual network that was not initially part of an existing community can learn the community's protocol with remarkable ease. Finally, we study, both qualitatively and quantitatively, the properties of the emergent protocol, providing some evidence that it is capturing high-level semantic features of objects.

## 1 Introduction

As state-of-the-art vision-processing deep networks start being deployed as components of real-life autonomous or semi-autonomous systems, the question arises of how such systems can communicate about the surrounding visual world, as seen through the lenses of their respective core visual components. Consider for example two industrial robots produced by different companies, or two self-driving cars from different makers, that need to coordinate about objects in their environment. One might be powered, say, by a ResNet convolutional net (He et al., 2016) trained on the ImageNet visual recognition challenge (Russakovsky et al., 2015), whereas the other might use a Vision Transformer trained with a self-supervised algorithm (Dosovitskiy et al., 2021; Caron et al., 2021). The set of object labels known to the two nets might differ (if labels are present at all), and, even when they coincide, the underlying categorization algorithms of the two nets might label the same object in different ways. Moreover, the original label set might not suffice to discriminate objects in the environment. For example, both nets might have been trained to categorize certain objects as *dogs*, but the current situation requires them to distinguish among multiple *dog* instances, or to denote an unfamiliar mammal neither network encountered at training time. Next, consider the scenario where a third system, relying on yet another visual network, is added to an existing "community" of systems that are already able to communicate. Do we need to develop a new communication protocol, or can the new agent quickly adapt to the one already in use? Could we eventually derive a universal protocol that any network-powered system could rapidly acquire? Note that, in the scenarios we are considering, the internal weights of the visual components might not be accessible for fine-tuning the network community as a single system, and, even if they were, this might be extremely costly or undesirable, as the core functionalities of the networks should not be altered.

We begin to tackle the challenges we outlined by studying whether a set of diverse pre-trained visual networks, augmented with a light interface layer, can develop a shared protocol when faced with the core communicative task of *referring to a specific object* among a set of candidates (Skyrms, 2010). We are inspired by recent work on *deep net emergent communication* (Lazaridou & Baroni, 2020). However, we adopt a different perspective with respect to this line of research. Instead of designing *ad-hoc* architectures specifically optimized on the

communication task, we look at emergent referential communication in communities of multiple pre-trained networks with heterogeneous architectures and training histories.

After reviewing related work (Section 2) and presenting our general setup (Section 3), we delve into our experiments in Section 4. First, in Section 4.1, we show that it is indeed possible for sets of heterogeneous pre-trained networks to successfully converge on a referent through an induced communication protocol. In Section 4.2, we study *referential generalization*, showing that the developed protocol is sufficiently flexible that the networks can, to some extent, use it to refer to objects that were not seen during the training phase (the "unfamiliar mammal" case we discussed above), as well as to tell apart distinct instances of the same object (the "distinguishing between different dogs" case above). In Section 4.3, we consider instead *agent generalization*. We show that, given a community of networks that have been trained to communicate, it is faster for a new pre-trained network with a different architecture, training objective or training corpus to acquire the existing community protocol, than for the enlarged community to develop a new protocol from scratch. This suggests that we could use a small set of networks to develop a "universal" protocol, that can then be taught in a supervised way to new networks that require the envisaged communication functionality. In Section 4.4 we offer some qualitative and quantitative insights into the protocol, suggesting that the network populations have developed a single shared code, that is not (only) referring to low-level image features, but also to higher-level semantic concepts. The Conclusion (Section 5) takes stock of what we found, and discusses the way ahead.

## 2  Related work

**Deep net emergent communication**  While there is a long tradition of work on communication emergence in communities of generic computational agents (e.g., Cangelosi & Parisi, 2002; Christiansen & Kirby, 2003; Wagner et al., 2003; Nolfi & Mirolli, 2020; Steels, 2012), there has recently been interest in the specific topic of *deep-net-based* emergent communication, with the aim to develop a protocol for autonomous information exchange between modern deep neural networks (Lazaridou & Baroni, 2020).

Since the ability to refer to a specific object in a shared environment is seen as one of the core building blocks of communication, many studies in this area have focused on the *referential game* setup, where a sender network must send a message to a receiver network to help it discriminate a target object among a set of candidates (e.g., Lazaridou et al., 2017; Havrylov & Titov, 2017; Dessì et al., 2021).

Inspired by human language, most work focusing on the referential game has been exploring a *discrete* communication setup in which the sender produces a symbol or a sequence of symbols. On the other hand, a continuous channel is often preferred in scenarios where agents have to tackle navigation-like tasks in environments featuring relatively few distinct referents (e.g., Foerster et al., 2016; Sukhbaatar et al., 2016; Tieleman et al., 2018; Kim et al., 2019; Singh et al., 2019). Carmeli et al. (2022) have recently shown that continuous communication outperforms discrete communication in the context of the referential game. As discrete communication imposes a non-differentiable bottleneck requiring special optimization techniques, we focus on the more natural continuous setup. We do however explore the effects of introducing a discrete channel in Appendix C.

Larger human speaker communities lead to the emergence of better-behaved protocols (e.g., Raviv et al., 2019). Many studies have assessed the impact of community size in deep network language emergence (e.g., Tieleman et al., 2018; Cogswell et al., 2019; Graesser et al., 2019; Li & Bowling, 2019; Ren et al., 2020). It appears that community-based training leads to desirable properties such as systematicity and ease of learning, at least when relatively small populations of homogeneous agents are considered. However, Rita et al. (2022) and Chaabouni et al. (2022) have recently shown that, in large-scale setups and when appropriate controls are considered, population training by itself does not bring about any special benefit. In our case, we are not so much interested in whether population-based training *per se* improves communication, but rather in whether it is possible at all to develop a shared protocol that is usable by multiple heterogeneous pre-trained networks, and easy to learn for new networks added to a community.

In typical population-based emergent communication studies, the networks in the population share the same architecture. Rita et al. (2022) looked at the effect of introducing sources of heterogeneity between senders

and receivers, finding that using agents with different learning rates enhances population-based protocols. While encouraging, this result was based on varying a single hyperparameter of otherwise identical networks, still far from our community-of-heterogeneous-networks scenario.

**Representation similarity and model stitching**  Our work touches upon the issue of whether different neural networks share similar representations (e.g., Li et al., 2016; Morcos et al., 2018; McNeely-White et al., 2020). Some of the work on this topic adopts a method known as "model stitching," which consists in connecting intermediate layers of two models to measure their compatibility (Lenc & Vedaldi, 2019; Bansal et al., 2021). This can be seen a form of model-to-model communication. Moschella et al. (2023); Maiorca et al. (2023), in particular, have recently introduced an unsupervised method that allows heterogeneous encoder and decoder networks to be "stitched" together in a 0-shot manner, based on second-order object representations (objects are represented by their similarities to a set of reference objects). Applying this approach to our referential communication scenario is a very exciting direction for future work.

**Self-supervised learning for image classification**  Various self-supervised discriminative training approaches have used objectives very similar to the one of our referential game, in order to derive classifiers without relying on annotated data (Chen et al., 2020; Caron et al., 2021), sometimes explicitly conceptualizing the setup as a multi-agent system (Grill et al., 2020). Some of the questions that are discussed in this field are also relevant for us. For example Lavoie et al. (2023) studied the effect of sparsifying representations into a constrained space, showing it improves generalization on new datatsets (we tried their methodology as an alternative way to derive discrete message representations in Appendix C, but found it not competitive in our setup). While there are many similarities between our core methodology and self-supervised learning, our aim is fundamentally different: we are not trying to induce an image classifier from scratch without annotated data, but to let distinct pre-trained networks (typically already trained as image classifiers) share information.

Our referential game is essentially a content-based image retrieval task, and in this sense our work is related to image retrieval using contrastive objectives (e.g., Wang et al., 2020b; El-Nouby et al., 2021). However, we do not train or fine-tune a full network to improve referent retrieval, but rather train shallow layers overlaid onto frozen networks with different architectures, and our emphasis, again, is on information sharing across networks.

**Multi-agent systems**  More broadly, our work fits into the wider area of multi-agent systems research, specifically cooperative learning and the decentralized POMDP setup (Panait & Luke, 2005; Oliehoek & Amato, 2016), a field in which there is also interest in the problem of optimizing the behavior of a diverse set of cooperating agents (e.g., Canaan et al., 2019). In the multi-agent decentralized learning domain, federated learning (e.g., McMahan et al., 2017; Lim et al., 2020) focuses on leveraging distributed model instances with different data sources to improve training. Like us, this domain focuses on communication efficiency and convergence of multiple agents (Wang et al., 2020a). However, we do not distribute training among a set of similar networks, focusing instead on multiple agents characterized by strong differences, and their ability to complete a communication task.

## 3  Setup

### 3.1  The referential communication game

Inspired by the core communicative task of *reference* (e.g., Skyrms, 2010), in the referential communication game a *sender* is given a target input (e.g., a picture) and issues a message (e.g., a word or, as in our case, a continuous vector). A *receiver* is exposed to a set of inputs, including the target, and must correctly point to the latter based on the message it receives from the sender.

In our context, the sender and the receiver are neural networks that include pre-trained and frozen visual modules, plus wrapper components in charge of the message-passing interface. We refer to the full interacting systems as *agents*. In each episode of the game, the sender agent receives a target image as input, processes it with its visual module, and outputs a message to the receiver. The receiver gets the message and is presented

Table 1: Pre-trained visual architectures employed

| Architecture | Type | Training | Parameters |
|---|---|---|---|
| **ResNet152** | CNN | Supervised ImageNet1k | 60.2M |
| **ResNet50** | CNN | Supervised COCO | 23.5M |
| **Inception** | CNN | Supervised ImageNet1k | 27.2M |
| **VGG 11** | CNN | Supervised ImageNet1k | 132.9M |
| **ViT-B/16** | Attention | Supervised ImageNet1k | 86.6M |
| **ViT-S/16** | Attention | Self-supervised ImageNet1k | 21M |
| **Swin** | Attention | Supervised ImageNet1k | 87.7M |

with a set of candidate images, containing the target and a series of distractors. For every candidate image, the receiver, using its visual module, constructs a representation, which is then compared to a representation of the received message. The receiver selects whichever candidate has the most similar representation to the message. If the selected image matches the target, the episode is successful. Success above chance level is only possible if the receiver learns to correctly interpret the message sent by the sender. Pseudocode for the referential game is presented in Appendix A.

Crucially, task accuracy only depends on target image matching, and does not require any kind of annotation. As such, the communication protocol is entirely learned in a self-supervised way.

Below, we refer to the basic setup, in which a single sender-receiver pair is jointly trained to play the game, as the *one-to-one* setting, further distinguished into *homogeneous* and *heterogeneous* cases, depending on whether the two agents use the same or different visual modules for image processing. To extend the game to the *population* setting, given a population of $N$ senders and receivers, we randomly sample one sender and one receiver at each step to play the game as previously described.

## 3.2 Agent architectures and training

As mentioned, in our setup both sender and receiver are neural networks made up of a frozen vision module and light feed-forward layers used for communication (Fig. 1).

### 3.2.1 Pre-trained vision modules

As vision modules, we use widely used networks that have freely downloadable parameters (Table 1). We mainly consider architectural variations of networks trained with a supervised classification objective, as this is the most common axis of variation across modern models. In particular, we use both convolutional (*ResNet*: He et al. (2016), *Inception*: Szegedy et al. (2014), *VGG*: Simonyan & Zisserman (2015)) and attention-based networks (*ViT*: Dosovitskiy et al. (2021), *Swin*: Liu et al. (2021)), trained on ILSVRC2012 ImageNet data (Russakovsky et al., 2015). We however also include in our collection a ResNet (from Desai & Johnson (2021)) that was trained on a different dataset, namely COCO (Chen et al., 2015). The latter involves fewer labels as well as stronger data imbalance (Gauen et al., 2017). Finally, we include an attention-based network, DINO (Caron et al., 2021), that was trained on ImageNet, but using a self-supervised objective (denoted by its ViT-S/16 architecture in Table 1).

The weights of the visual models are frozen (in blue in Fig. 1) and they are not modified at any point during our experiments. As output from the vision modules, we use their last layer, before the Softmax non-linearity is applied. Therefore, the vision module takes as input images and outputs vector representations of the images.

While we are interested, by problem definition, in communication between *pre-trained* networks, whose visual processing abilities will largely derive from the pre-training phase, in Appendix B we try to assess to what extent the discrimination abilities that we'll see emerge in the agents are a simple by-product of pre-training. The results there suggest that this is not the case.

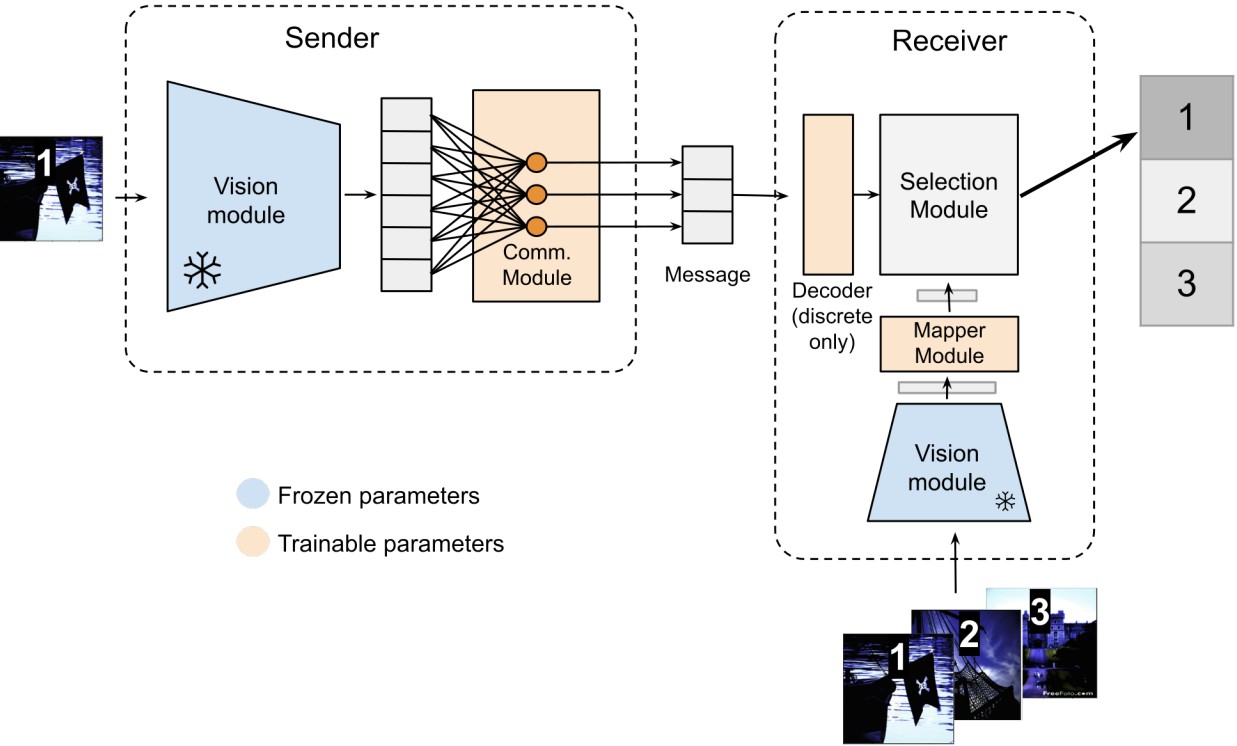

Figure 1: Referential game setup and agent architectures. A target image is input to the sender, that extracts a vector representation of it by passing it through a pre-trained frozen visual network. This vector representation is fed to the feed-forward *Communication Module*, that generates a message consisting of another continuous vector, that is passed as one of the inputs to the Receiver. The receiver also processes each candidate image it gets as input by passing it through a pre-trained frozen visual network (which can have a different architecture from the one of the sender), obtaining a set of vector representations. These are fed to a *Mapper Module*, another feed-forward component that maps them to vectors in the same space as the sender message embedding. The *Selection Module* of the receiver simply consists of a parameter-free cosine similarity computation between the message and each image representation, followed by Softmax normalization. The receiver is said to have correctly identified the target if the largest value in the resulting probability distribution corresponds to the index of the target in the candidate array. Note that no parameters are shared between sender and receiver (except those of the frozen visual modules in the case in which the two agents are using homogeneous visual architectures).

### 3.2.2 Trainable communication components

The second part of an agent's architecture consists in the layers devoted to handling communication. In both sender and receiver, all communication-related parameters are trainable, as shown in orange in Fig.1.

In the sender, the communication module is a small linear feed-forward layer, with between 6k and 264k parameters. For comparison, the frozen vision modules in Table 1 all have more than 20M parameters. The communication module takes as input the image representation, and outputs a continuous vector we refer to as the *message* emitted by the sender. We ran a hyper-parameter search for message dimensionality (Appendix D). Since we found a ceiling effect, we decided to run all experiments with both the smallest (16) and the largest (64) size that reached peak validation performance.

Note that, when we use the term *communication protocol* below, we refer to the deterministic input-conditioned message-emitting policy the sender converges to after training.

In the receiver, the vision module is followed by a mapper module. Like the sender's communication module, it maps image representations to a new compressed state. The message from the sender is compared to these image representations. Following standard practice in the emergent communication literature, the comparison is conducted using the parameter-free cosine similarity measure. The receiver mapper module is larger than the sender communication component, being composed of two linear layers with a hidden dimension, as we adopted the architecture of Dessì et al. (2021) without further tuning. The number of parameters remains well below that of the pre-trained vision modules. As an example, VGG (132.9M parameters) is the vision module that has the largest output dimension, and therefore requires the largest mapper module, with 8.4 million parameters.

Emergent communication research often adopts a discrete channel, as the latter more closely emulates human linguistic communication. We experimented extensively with a discrete interface, as reported in Appendix C. However, as discrete communication does not lead to clear advantages over continuous communication, and it involves a more involved and brittle optimization process, we focus here on the continuous interface we just described.

### 3.2.3 Training

As the sender-receiver interface is continuous, the whole system can be straightforwardly trained by back-propagating the standard cross-entropy loss, obtained by comparing the receiver output to the ground-truth target.

Batch size is set at 64, the largest value we could robustly fit in GPU memory. As we sample distractors directly from training batches, on each training step the referential game is played 64 times, once with every different image in the batch as target and the other 63 images serving as distractors. All experiments are implemented using the EGG toolkit (Kharitonov et al., 2019). All parameters needed to reproduce our experiments with the toolkit (both the continuous setup reported here, and the discrete one discussed in Appendix C) can be found in Appendix E. Evidence of training convergence is provided in Appendix F in terms of gradient evolution profiles. Appendix G reports results for variants of training in which we used image augmentations and hard distractors.

### 3.3 Datasets

As nearly all agents rely on vision modules pre-trained on the ILSVRC2012 training set, we sample images from the validation data of that same dataset (50,000 images) to teach them to play the referential game, while reserving 10% of those images for testing (note that we do not use image annotations). We emphasize that our *ImageNet1k* communication training and testing sets are thus both extracted from the original ILSVRC2012 *validation* set.

Agents are also tested on an out-of-domain (*OOD*) dataset containing classes from the larger ImageNet-21k repository, as pre-processed by Ridnik et al. (2021). We selected the new classes among those that are neither hypernyms nor hyponyms of ImageNet1k classes, and we made sure that no OOD class had a

Table 2: Percentage *accuracy* and *learning time* of agents playing the referential game. Learning time counts the number of epochs until maximum accuracy is reached (bigger numbers indicate slower convergence). Results are reported for two message dimensions. The *homogeneous* row reports mean scores and standard deviations across 7 pairs of same-architecture nets, each trained separately. The scores in the *heterogeneous* row are averaged across all 42 possible different-architecture pairings, each trained separately. Population accuracy is averaged across all 49 possible net pairings after joint training. There is no standard deviation around population learning time as we have a single jointly-trained population.

| | Accuracy (16) | Learning Time (16) | Accuracy (64) | Learning Time (64) |
|---|---|---|---|---|
| **Homogeneous** | $99.3 \pm 0.8$ | $9 \pm 8$ | $98.6 \pm 2.9$ | $6.5 \pm 8.1$ |
| **Heterogeneous** | $95.3 \pm 3.4$ | $7 \pm 3$ | $99.2 \pm 1.1$ | $9.7 \pm 6.5$ |
| **Population** | $96.8 \pm 2.7$ | $28$ | $98.2 \pm 2.2$ | $24$ |

WordNet path similarity score[1] above 0.125 with any ImageNet1k class.[2] We moreover used a number of heuristics, combined with manual inspection, to make sure the new classes were more or less at the same level of granularity as the ImageNet1k categories. After applying these filters, we were left with 52 classes denoting objects that belong to semantic domains disjoint from those represented in ImageNet1k, such as trees (*olive tree*, *fir*, *red pine*) and human professions (*carpenter*, *physician*, *organ grinder*). For each of them, we sampled all images available in the pre-processed ImageNet-21k dataset of Ridnik et al. (2021) (guaranteed to be at least 450). We further split the OOD data into a larger 90% set that is used for testing OOD communication accuracy (Section 4.2 below). We additionally train classifiers on this 90% partition for the experiments reported in Appendix H, using the remaining 10% of the OOD data to test them.[3]

## 4   Experiments

We will start our experimental report by comparing single agent pairs and larger communities in the straightforward setup in which test images depict new instances of referents used for communication training (Section 4.1). We will then look at protocol generalization along two axes: communicating about new *referents* (Section 4.2) and communicating with new *agents* (Section 4.3).

### 4.1   Referential communication of homogeneous and heterogeneous networks

Starting from the baseline scenario in which there are only one sender and one receiver with identical visual modules (one-to-one *homogeneous* setting), we look at the effect of pairing two *heterogeneous* architectures (one-to-one *heterogeneous* setting), and then scale up to training a full population of agents at once (*population* setting).

The results are presented in Table 2. We observe first that it is possible to quickly train two homogeneous agents to near perfect referential accuracy (first row). Importantly, as shown in the second row of the table, **pairing agents with different visual architectures only weakly affects accuracy and learning time, if at all**. We conjecture that this ease of training in the heterogeneous setup is at least partially due to the fact that different architectures must ultimately extract comparable semantic features from images during pre-training, facilitating the task of making them communicate (although, as we will see below, there is no strong correlation between similarity of pre-trained visual module representations and communication performance). Detailed performance across all tested agent pairs is available in Appendix I, and we comment on some general patterns observed there in what follows. We focus here on agents trained with 16-dimensional messages, as with the 64-dimensional channel we are uniformly near the ceiling.

Communication is easy for pairs that have different architectures. For example, communication between a ViT sender and a VGG receiver reaches 95% accuracy in 4 epochs. Communication is also easy for visual

---

[1]As implemented in the NLTK toolkit: `https://www.nltk.org/`.

[2]The COCO dataset, used to pre-train one of our visual modules, contains four labels slightly above the threshold, with maximum similarity at 0.17.

[3]We provide scripts to reproduce our ImageNet1k and OOD datasets at `https://github.com/mahautm/emecom_pop_data`

modules based on different training objectives: for example, ResNet-to-DINO communication reaches 98% accuracy in 12 epochs. It might look like communication is harder for pairs trained on different corpora, as 4/5 pairs requiring 10 or more epochs to train involve ResNet-COCO as the receiver agent. However, the latter also requires 29 epochs to converge in the homogeneous setup, suggesting that the (relative) issue with this module is the mismatch between its pre-training corpus (COCO) and the one used for communication training (ImageNet1k), rather than its heterogeneity with respect to the other agents.[4]

**It is also possible to train the full agent population to very high accuracy in relatively few epochs** (*Population* row in Table 2). In this setup, we use all 7 modules as both senders and receivers, randomly pairing them at training time, ensuring each agent interacts with every other agent. A common communication protocol emerges that is mutually intelligible across all 49 agent pairings. Note that the large increase in learning time compared to the one-to-one setups is only apparent, because during population training each agent is only seeing 1/7 of the data in each epoch.[5] Thus, in terms of actual data exposure, a joint count of 28/24 epochs correspond to 4/3.3 effective epochs of data exposure for a single agent, on average. Convergence is thus faster on average in population training than in one-to-one training.

We investigated whether the (generally small) differences in accuracy or learning time between agents with different visual modules is accounted for by the underlying discrepancy between their pre-trained visual representations. As different visual modules have different output dimension, we used RSA (Kriegeskorte et al., 2008) to measure the correlation between visual module output similarity and accuracy/learning time in communication training (measured for population pairs and one-to-one couplings, respectively). We did not find a significant correlation in either case, suggesting that communication training is quite robust to underlying visual representation differences.

## 4.2 Referential generalization

Having established with the previous experiment that it is possible to train a light communication layer that allows networks with heterogeneous architectures to successfully agree on a target referent, we go back to the questions about practical referent generalization we asked in the introduction. First, we explore whether the protocol can make more granular distinctions among objects coming from the same ImageNet1k class that were used to develop it. For these purposes, we built a special test set where *all* distractors are from the same ImageNet1k class as the target (so that, say, one target magpie needs to be distinguished from other magpies, instead of swans and bullfrogs). We tested all 1,000 ImageNet1k classes in smaller batches of 32 candidates, to ensure we had enough images in all cases. In this *single-class* setup, randomly choosing an image would put baseline accuracy at around 3.1%. Note that, since we are interested in out-of-the-box generalization, we use, zero-shot, the very same protocols induced with random distractors in the experiments above, without any special adaptation to the single-class setup.

Table 3 shows that **communication is at least partially successful at a more granular level than ImageNet1k classes**, despite a large drop from the random-distractor performance reported above. There is a noticeable discrepancy between 16- and 64-dimensional messages, with the latter being generally better than the former, especially in the heterogeneous setup. We find this result intriguing, as we would have expected smaller message dimensionality to act as a regularizing bottleneck, leading to better generalization performance.

Next, we ask whether a communication protocol can also generalize, zero-shot, to referents that were not seen at training time. In particular, the systems trained on the referential game using ImageNet1k data are now tested on the OOD dataset, that is entirely composed of new-class objects (see Section 3.3 above).

As Table 3 shows, there is a significant accuracy drop (larger in the more challenging heterogeneous and population setups), but performance remains good enough to suggest that **the protocol is at least to some extent able to generalize to images depicting referents not seen during training** (recall

---

[4]The slightly larger average learning time and corresponding standard deviation in the homogeneous setup compared to the heterogeneous one are accounted for by the extra time it takes to train the ResNet-COCO sender-receiver pair. Note that, during heterogeneous training, at least one of the two agents is always ImageNet-based.

[5]In other words, Table 2 is comparing on the same scale the time it takes to train 2 agents in the one-to-one setups to the time it takes to train 14 agents in the population setup.

Table 3: Percentage accuracy on single-class and Out of Domain sets for both message dimensions.

|  | Single-class (16) | OOD (16) | Single-class (64) | OOD (64) |
|---|---|---|---|---|
| **Homogeneous** | $47.2 \pm 3.4$ | $90.5 \pm 8.0$ | $47.4 \pm 6.4$ | $94.8 \pm 10.4$ |
| **Heterogeneous** | $31.0 \pm 22.2$ | $62.9 \pm 17.6$ | $48.6 \pm 2.1$ | $82.2 \pm 13.8$ |
| **Population** | $34.7 \pm 6.8$ | $72.2 \pm 13.8$ | $36.5 \pm 8.6$ | $72.2 \pm 44.8$ |

that chance performance is here at $1/64 \approx 1.6\%$). Larger dimensionality seems again to help generalization, at least in the homogeneous and heterogeneous setups.

### 4.3 Agent generalization

Having tested how well the protocol can be applied to new *referents*, we now consider whether it is possible to efficiently extend it to new *agents*, as in the practical scenario where a new autonomous device is added to an existing network of such items. In particular, we ask whether a new agent joining a community of agents already trained to communicate with each other can efficiently learn their communication protocol.

To test ease of learning of a communication protocol, we add a single network, which we call the *learner* agent, to an existing community, either as sender or as receiver.[6] This new agent has to play the referential game with the original agents and learn their protocol. To single out ease of learning of the communication protocol itself, and not the original agents' adaptability, we freeze all communication layer parameters except for the learner agent's. The new communication layer is therefore the only one being updated. The focus of this experiment is on the heterogeneous learners, where the new agent's vision module is different from that of the original agents. We nonetheless observe that, in the homogeneous case (not reported here), communication protocols are learned very quickly, needing at most one epoch to reach perfect or near-perfect accuracy.

In the (heterogeneous) one-to-one setup, we train a communication protocol for each of the 49 possible different architecture pairs.[7] For each pair, we then add a learner sender or a receiver for every vision module not in the original pair (e.g., if the protocol was trained on the ResNet152-Swin pair, we test, as both learner senders and receivers, Inception, VGG 11, ViT-B/16, ViT-S/16-DINO and Resnet50-COCO). In the population setup, the 7 possible leave-one-out populations of 6 different senders and 6 different receivers are trained on the referential game, and, for each, the learner agent will use the 7th vision module that was left out, acting as either sender or as receiver.

Test accuracy and learning speed of the learner agents are reported in Fig. 2. In all runs with a 64-dimensional communication channel, **the learner agent succeeds at learning the communication protocol** with accuracy of at least 90%. There is some more variance with 16-dimensional messages, but in this case as well nearly all runs converge to very high accuracy. Receivers learn better and faster on average (94%/95% for one-to-one with 16/64-messages, respectively, and 96%/97% for population) than senders (88%/95% and 95%/97% respectively).

The comparison between the one-to-one and population setups reveals no clear winner: population accuracy is on average higher than one-to-one accuracy with 16-dimensional messages but lower with 64-dimensional messages. Population accuracy, however, is more stable in both setups, showing very little variance from case to case.

Regarding learning speed, **agents that learn a pre-existing communication protocol reach high accuracy faster than it would take to retrain the whole population from scratch**, as illustrated by the population-from-scratch baseline in Fig. 2. The communication protocol is indeed initially very easy to learn for a new agent. There is no clear difference in learning speed between the one-to-one and population setups.

---

[6]Alternatively, a single pair of agents could initially be trained, and then all other agents could be sequentially added to the growing community. As we obtain comparable results when adding each learner in turn to all possible network pairs or sextuples, we reasonably expect that this alternative setup would also lead to similar outcomes.

[7]These pairs also include homogeneous senders and receivers, but we will always add an heterogeneous agent to the pair.

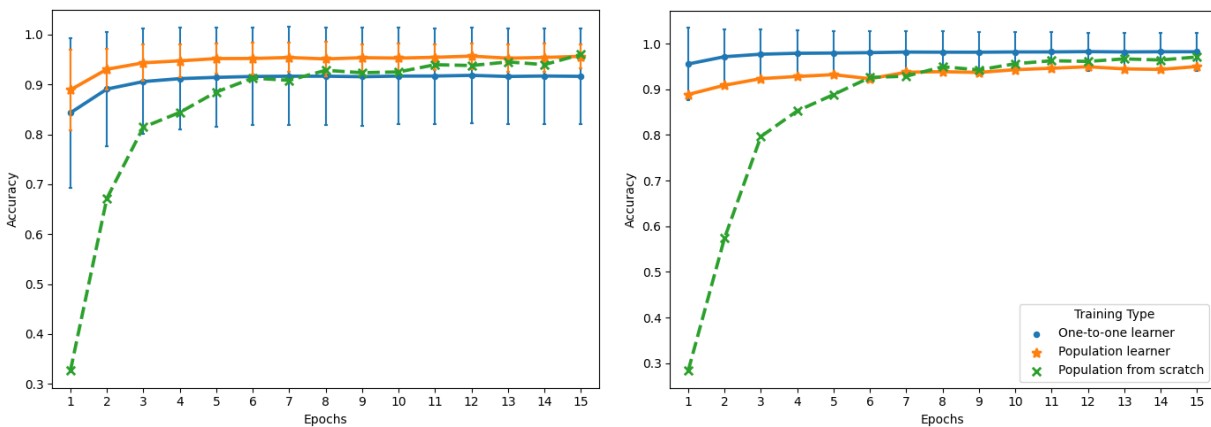

Figure 2: Test accuracy and learning speed of learner agents (left: 16-dimensional messages; right: 64-dimensional messages). Blue line: learning curve on test data for learner agent added to a communicating pair, averaged across all possible heterogeneous triples. Orange line: learning curve for learner agent added to an existing community, averaged across all possible leave-one-out cases. Vertical bars indicate standard deviation across cases. As a baseline for learning speed, the dashed green line shows the learning curve when training the whole 7x7 population at once from scratch.

### 4.4 Protocol analysis

Having shown its effectiveness, we turn now to a qualitative and quantitative examination of the emergent protocol. For this analysis, we focus on the overall better-performing 64-dimensional message setup.

To start, some qualitative sense of what messages are expressing is conveyed by the examples in Figure 3 (further examples shown in Appendix J). These are representative cases of when the VGG 11 sender + ViT-B/16 receiver pair that was trained in the population setup failed to converge on the target. In a large majority of them, the wrongly selected image is depicting an entity from the same class as the target (first block of examples in the figure). The fact that often the relation is conceptual rather than perceptual (as in the second example of Figure 3, where the target is a drawing of a rocket, but the receiver picked a photograph of a rocket) suggests that agents are communicating about abstract categories, rather than low-level visual features. Of the minority of cases in which the wrongly selected image is not from the same class, some errors are relatively transparent, such as the ones in the second block of Figure 3, where we see similar shapes and textures in the target and in the wrongly picked distractor. Still, outside the frequent same-class errors, the most common case is the one illustrated by the last block of the figure, where the nature of the error is not clear. Such cases call for a more quantitative investigation of the protocol, which we will now discuss.

We begin such investigation by applying the "Gaussian blobs" sanity test introduced by Bouchacourt & Baroni (2018). We check whether the agents trained to communicate on the ImageNet1k dataset are able, at test time, to play the referential game on blobs of Gaussian noise instead of genuine images. Success would mean the agents developed degenerate communication strategies, focusing on pixel-level image features that can transfer to the task of discriminating between blobs of Gaussian noise, rather than on more desirable high-level semantic information. We find that all one-to-one- and population-trained pairs are indeed at chance level on Gaussian blobs, suggesting they are not just communicating about low-level image features.

To investigate further what image features are transmitted by the communication channel, we perform a set of perturbations affecting various aspects of the input images (an analysis of perturbations of the *communication channel* is reported in Appendix K). The effects of the perturbations on an example image are shown in Fig. 4. We perturb images from the ImageNet1k dataset that were seen by the agents during referential game training, where the receivers have perfect accuracy. In each experiment, we apply the same transformation to all images, both those seen by the sender and those seen by the receiver. Note that we test

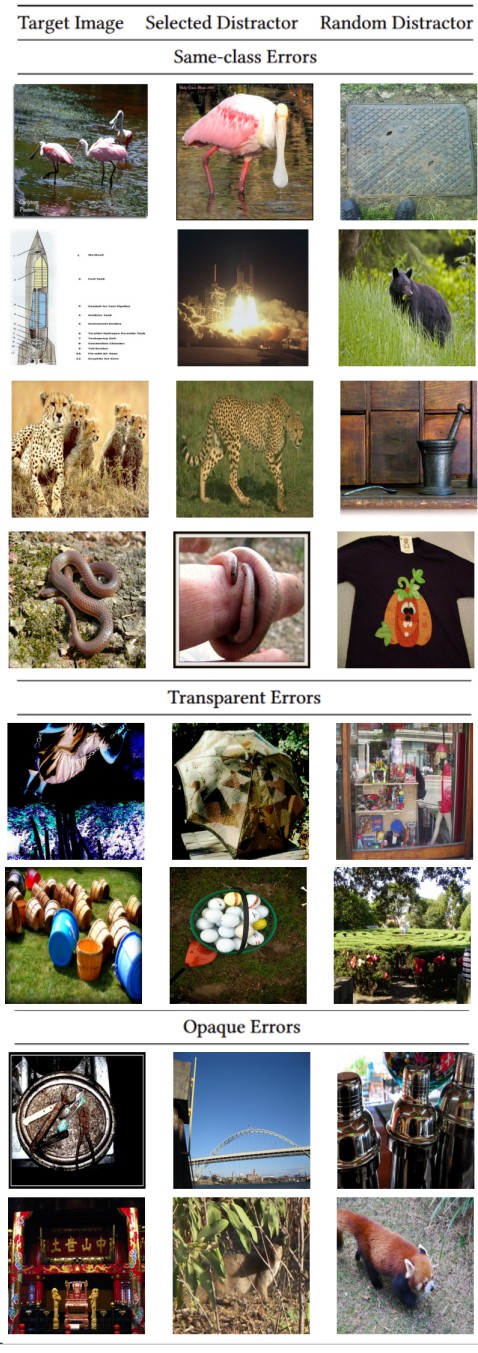

Figure 3: Examples from the ImageNet1k-based test set of mistakes made by a VGG 11 sender + ViT-B/16 receiver pair trained in the population setup. We also show an additional random distractor from the batch for context.

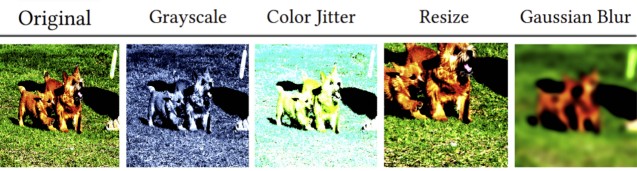

Figure 4: Perturbation examples

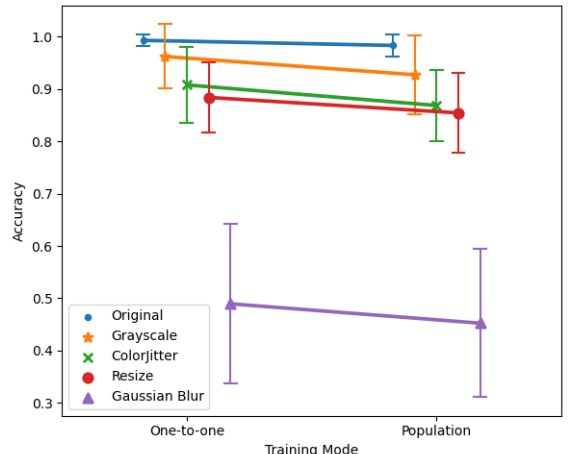

Figure 5: Accuracy after various image perturbations for the one-to-one and population setups. Gaussian blur was uniformly sampled within the [0.1,10] range. Mean accuracies and corresponding standard deviations are given across all 36 possible agent pairs. We add a small horizontal offset to the accuracy values for different perturbations to make them all visible. The horizontal lines connecting one-to-one and population accuracies for the same perturbation are meant to ease comparison between the two settings. Random baseline is 1.6% (randomly selecting one of the 64 images).

senders and receivers that have been trained with non-corrupted images, and only face the perturbations at test time.

Results for both the one-to-one and the population setups are reported in Fig. 5. All ablations affect accuracy. However, the color ablations (transforming the image to *grayscale* and *jittering colors*) and *resizing* the image are having a much smaller impact on performance than adding *Gaussian blur*. This is in line with our intuition about how much these changes should affect the ability to decode the contents of an image. As the example in Fig. 4 shows, color and size ablations don't normally affect the human ability to discriminate the object depicted in the image, whereas blurring the image can easily make its contents difficult to decode for us as well.

We finally observe that all perturbation types affect one-to-one and population-trained agents in a similar manner, with population-trained agents being marginally more affected.

Still, an important distinction in favor of population-based training is that in this setup, as shown in Fig. 6, the senders converge to using very similar messages to denote the same image, approximating a single protocol. Note that the observation is not trivial: the senders could in principle have developed very different protocols, with the receivers learning to process the union of these distinct protocols (we informally observed something of the sort taking place in the emergence of discrete population-based communication). Fig. 6 also shows that the distance between one-to-one senders, that have been trained independently of one another, is significantly larger on average than that between population-trained senders, and virtually indistinguishable from the baseline of averaging distances between messages representing different images for a single sender.

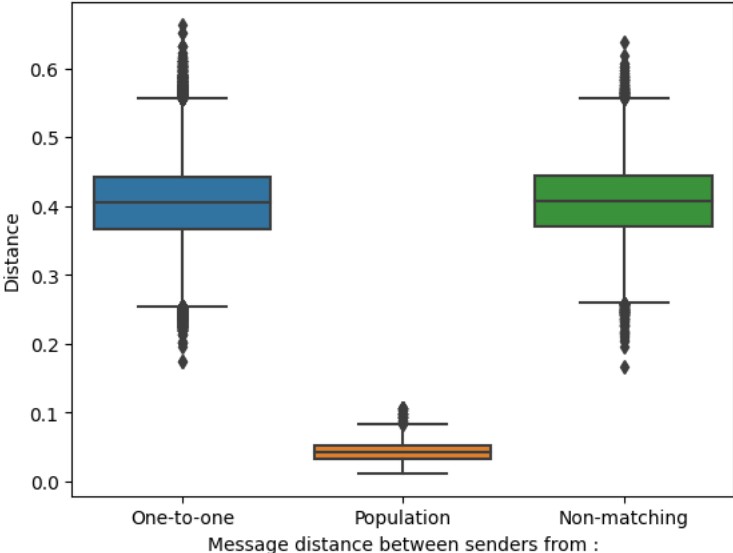

Figure 6: Distributions of cosine distances between messages sent by two different senders for the same ImageNet1k training set images. The different senders are either trained in the heterogeneous one-to-one setup (blue) or in the same population (orange). As a baseline, the green boxplot shows distances between the messages produced for *different* input images by population-trained VGG 11 and ViT-B/16 senders.

To further illustrate, and build upon, the observation that population training has converged to a single protocol, in Appendix H we report an experiment in which the common representations induced in this setting are used for zero-shot classifier transfer.

## 5 Conclusion

Coming back to the questions we asked in the Introduction, we found that it is possible for agents based on pre-trained deep visual modules with different architectures, training objectives and training data, to induce a successful referential communication protocol through a self-supervised training procedure that only involves a light communication layer, without touching the visual module weights. The architectural extensions are simple and straightforward, suggesting that differing underlying pre-trained visual models already converged to similar representations.

The networks can use the emergent protocol, to some extent, to communicate about referents that are only distinguishable at a more granular level than used during communication training; and they can, to some extent, communicate about referents coming from categories that were not seen during pre-training nor in the communication protocol induction phase.

Importantly, we showed that a new deep net agent can successfully learn the protocol developed by an existing community, and that it can do so in less time than it would take to re-train the enlarged community including the new agent from scratch. This points to the possibility of developing a "universal" deep net protocol, that could be quickly taught to any new model before it is deployed in multi-network communication scenarios. We further found that the emergent protocol possesses a number of intuitive characteristics suggesting that it is focusing on semantic properties of the objects depicted in images, rather than on low-level visual features.

Despite our encouraging set of initial results, clearly much work is still needed before the envisaged communication layer can be of practical use. First, we observed a drop in performance in the referent generalization experiments compared to communication about in-domain referents. Future work should further explore the generality of the protocol, and devise ways to broaden it. An intriguing direction for future work would be to include visual encoders trained on objectives that involve mapping images to natural language (e.g.,

Radford et al., 2021), as the representations of such encoders might embed positive biases inherited from language.

We have motivated our study in terms of nets embedded in robots and self-driving cars. Such systems will not perceive the world as a discrete set of pictures, but rather as a fully embodied egocentric image flow. This scenario obviously poses further challenges. For example, two agents that need to communicate about an object would almost never have an identical view of it. How our approach could scale up to this setup is another important direction for future work.

Referent denotation is an important building block of communication, but it's clearly just the starting point, and further work should extend deep net communication to other functions of language, such as highlighting different properties of objects, describing actions, and distinguishing between providing information, asking for it and giving instructions. These features will in turn require extending the setup to multiple exchanges, and to agents that can function both as senders and as receivers.

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

## A    The one-to-one referential game in pseudocode

```
initialize sender and receiver
for n training iterations do
    for batch of images b in training data do
        recdists = list()
        groundtruths = list()
        for image i in b do
            groundtruths.add(one-hot vector with 1 at i's index)
            sender receives i
            sender outputs message vector
            receiver receives message and b
            receiver processes message and each image in b to generate representations
            recdist = cosine between receiver representations of message and images in b
            recdist = softmax(recdist)
            recdists.add(recdist)
        end for
        compute cross-entropy loss CE between groundtruths and recdists
        backpropagation + update
    end for
end for
```

For the $B$ images in a batch, the cross-entropy loss function is computed as follows:

$$\text{CE} = -\sum_{i=1}^{B} \sum_{j=1}^{B} recdists_{ij} \log(groundtruths_{ij})$$

In the population-based setup, all available senders and receivers are initialized at the beginning, and a sender and a receiver are randomly selected at each iteration of the second *for* loop.

## B    Using visual representations without a communication layer

Our setup assumes that we need to connect networks that have already been pre-trained on visual tasks. While our main objective is to evaluate communication success, an interesting question is to what extent the communication layer is adding something non-trivial to what is already encoded in the pre-trained visual representations of the networks. To partially assess this, we attempted to directly play the game with "agents" that generated messaged directly from last-layer visual representations. For each network, we clustered all images using k-means, setting $k$ to the ground-truth number of classes in the dataset. We then treated the k-means clusters as the "message" produced by the sender. The receiver would then randomly pick an image belonging to the correct cluster from the candidate set.

As table 4 shows, in the homogeneous case where sender and receiver have the same architecture,[8] k-means pairs achieve on average ImageNet1k and OOD performance well above chance level, but clearly lagging behind the networks that underwent communication training (performance $> 98\%$ and $> 90\%$ on these tasks, respectively). This confirms that the latter has enriched the networks with discrimination abilities that go beyond what's directly inherited from (mostly) supervised pre-training.

---

[8] For the heterogeneous case, where sender and receiver are different networks, we tested various ways to align the clusters on the training set, but performance was consistently scarcely above chance.

Table 4: Percentage accuracy of agents evaluated on the ImageNet1k and OOD sets. Agents are unsupervised clusters of visual representations built using the k-means algorithm.

|  | ImageNet1k | OOD |
| --- | --- | --- |
| **Homogeneous** | $49.8 \pm 17.9$ | $21.2 \pm 7.3$ |

Table 5: Percentage accuracy and learning time of agents playing the referential game with discrete messages once both agents have reached convergence. Terminology as in Table 2.

|  | **Accuracy** | **Learning Time** |
| --- | --- | --- |
| **Homogeneous** | $78.2 \pm 0.1$ | $20 \pm 1$ |
| **Heterogeneous** | $71.1 \pm 4.3$ | $22 \pm 2$ |
| **Population** | $62.4 \pm 3.2$ | 23 |

## C  Discrete experiments

Building on intuitions from the deep net emergent communication literature, we studied whether imposing a discrete bottleneck would lead to a better, more general protocol than direct communication through a continuous channel. In a first experiment, we implemented a discrete bottleneck letting the agents exchange a one-hot vector, by optimizing it through the widely used Gumbel Softmax differentiable relaxation (Havrylov & Titov, 2017; Jang et al., 2017; Maddison et al., 2017), fixing vocabulary size (dimensionality of the message vector) to 256. Note that, when communication is discrete, a further decoder component is added to the receiver architecture, mapping the discrete messages back to the continuous space in which it can be compared to the candidate image representations.

We did not find any evidence that the discrete bottleneck helps. Not only is the discrete protocol significantly less accurate than the continuous one (Table 5), but this low performance hides any trace of better generalization capabilities (Table 6).

When generalizing a discrete communication protocol to a new agent (Figure 7), as done for continuous messages in Section 4.3, we observe convergence to a lower accuracy at a slower pace. Still, this setup confirms even more dramatically that less time is required to train a new agent to top accuracy than is needed to retrain a larger group from scratch. Specifically, the population trained from scratch would require 23 epochs to reach its best accuracy, whereas 5 epochs are sufficient to teach a new agent an already accurate protocol. The population training paradigm is more stable across architectures, as with continuous messages.

In order to make sure that our negative results concerning discrete communication are not due to an unfortunate choice of hyperparameters, we repeated most experiments across a range of settings. We report here results for the more interesting population setup, but similar patterns are observed in one-to-one training. Table 7 shows that there is a small increase in accuracy when enlarging vocabulary size (compared to the results obtained with 256-dimensional messages reported in tables 5 and 6 above), but performance is still well below that of the continuous case even when using a vector as large as 8192 dimensions (compared to 16 dimensions in the continuous case). Note that using vocabulary sizes below 256 (not shown in the table) led to failure to converge to non-random accuracy.

We also considered replacing Gumbel Softmax optimization with the commonly used policy gradient REINFORCE algorithm for fully discrete optimization (Williams, 1992). Gumbel Softmax (Table 7) proved to be the more effective training strategy. Switching to REINFORCE (Table 8) causes accuracy to drop to near-chance levels.

REINFORCE allows a straightforward implementation of multi-symbol autoregressive communication, where we let the sender and receiver produce/parse symbol sequences by replacing their communication encoding/decoding components with a recurrent LSTM network (Hochreiter & Schmidhuber, 1997). We set maximum message length to 10 symbols. Performance of this multi-symbol approach stays capped at 1.6% accuracy, which corresponds to the random baseline.

Table 6: Percentage accuracy on single-class and OOD test sets for discrete communication

|  | Single-class | Out of Domain |
|---|---|---|
| **Homogeneous** | $13.2 \pm 4.1$ | $43.2 \pm 5.0$ |
| **Heterogeneous** | $16.0 \pm 6.2$ | $29.1 \pm 4.3$ |
| **Population** | $8.9 \pm 2.1$ | $25.8 \pm 4.5$ |

Figure 7: Test accuracy and learning speed of discrete learner agents. Blue line: learning curve on test data for learner agent added to a communicating pair, averaged across all possible heterogeneous triples. Orange line: learning curve for learner agent added to an existing community, averaged across all possible leave-one-out cases. Vertical bars indicate standard deviation across cases. As a baseline for learning speed, the dashed green line shows the learning curve when training the whole 7x7 populations at once from scratch.

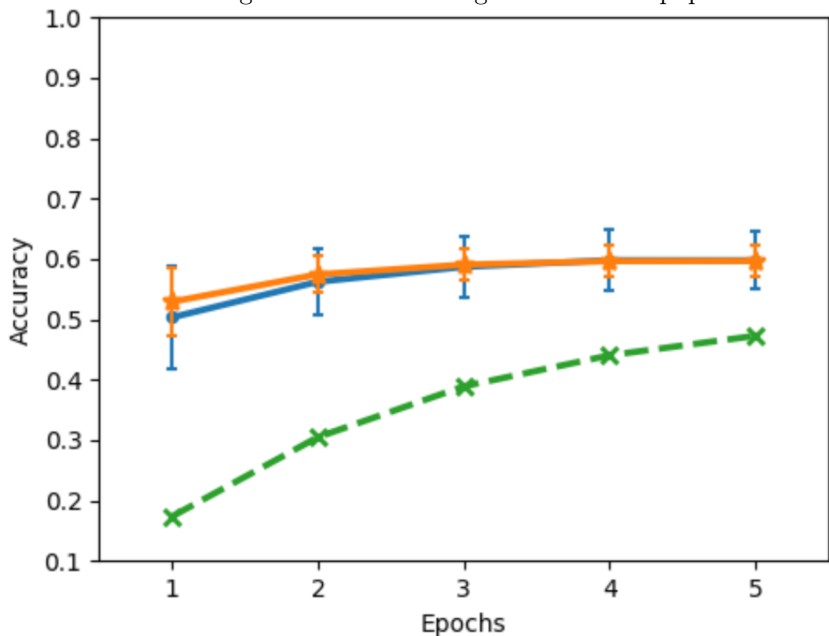

We conjectured that the overall disappointing performance of discrete communication is due to the difficulty of optimizing agent coordination when the error gradient cannot be passed through the discrete channel. We thus tried the *simplicial* embeddings from Lavoie et al. (2023), as this approach results in nearly discrete representations while not requiring discrete-channel optimization. We applied the method on top of the continuous linear layer, which maps the vision module representation space to message space (16 dimensions). We cut the 16 dimensions into 2, 4 or 8 groups of 8, 4 or 2 vectors respectively, and apply a softmax non-linearity within those groups, before concatenating everything back to 16 dimensions. The number of groups had very little impact on results, and led to the same average performance. Accuracy with both in-domain and

Table 7: Percentage accuracy of agents evaluated on the two ImageNet-based validation sets in function of vocabulary size. Agents were trained with Gumbel Softmax in a population setup.

| Vocabulary size | ImageNet1k | OOD |
|---|---|---|
| **512** | 62.1 | 26.8 |
| **1024** | 66.8 | 31.1 |
| **2048** | 71.7 | 32.1 |
| **4096** | 74.3 | 33.4 |
| **8192** | 78.5 | 35.2 |

Table 8: Percentage accuracy of agents evaluated on the two ImageNet validation sets. Agents were trained with REINFORCE in a population setup.

| Vocabulary size | ImageNet1k | OOD |
|---|---|---|
| **256** | 5.6 | 3.9 |
| **512** | 3.4 | 2.4 |
| **1024** | 3.3 | 2.1 |
| **2048** | 2.8 | 1.6 |
| **4096** | 2.4 | 1.8 |
| **8192** | 2.2 | 2.1 |

Table 9: Percentage accuracy of agents evaluated on the two ImageNet validation sets. Agents were trained with simplicial embeddings.

| | ImageNet1k | OOD |
|---|---|---|
| **Homogeneous** | $84.4 \pm 15.4$ | $63.9 \pm 17.7$ |
| **Heterogeneous** | $82.2 \pm 5.9$ | $45.6 \pm 12.3$ |
| **Population** | $62.7 \pm 3.4$ | $24.9 \pm 11.1$ |

out-of-domain datasets is reported in Table 9, and it remains below that of plain continuous communication, without the expected boost in generalization abilities.

If using a discrete channel is considered a priority (for example, because it might lead to a more interpretable protocol), other discrete optimization techniques could also be explored, such as more advanced policy gradient methods (Francois-Lavet et al., 2018), or quantization approaches such as the one recently proposed by Carmeli et al. (2022) (although we tried something similar, with less-than-encouraging results, in Appendix K).

## D   Hyperparameter search for continuous communication

Since, following standard practice, we are using the ILSVRC2012 validation partition for testing (the ILSVRC2012 test partition is not available), we rely on a separate dataset, namely CIFAR100 (Krizhevsky et al., 2009), for hyperparameter tuning.

The two main hyperparameters to consider in continuous communication are the size of the continuous message, and the non-linearity applied to it. We performed multiple grid searches to determine the best and most efficient combination, through referential game experiments run on CIFAR100. Table 10 shows accuracy at epochs 1 and 25 of an agent pair based on the Inception vision module.

On our validation set, all dimensions between 16 and 64 reach perfect accuracy. We thus decided to run the main experiments with both the smallest (16) and largest (64) dimensionalities in this range. We leave it to future work to explore even larger dimensionalities. We remark, however, that one of our desiderata is to keep the communication module small, and increasing dimensionality inevitably expands the size of the module. We also note here that dimensionalities smaller than 16 still manage to carry a lot of information. We are above the random chance level of $1/64 \approx 1.6\%$ even with 2 dimensions.

The sigmoid non-linearity has a small margin over softmax (clearer in the smaller dimensions, where there is no ceiling effect), and we thus pick the former for the main experiments.

## E   Experiment hyperparameters

All hyperparameters required to reproduce the continuous (main-text) experiments using the EGG toolkit are provided in Table 11. Hyperparameters for discrete experiments (Appendix C) are in Table 12. Table 13 has hyperparameters for the REINFORCE experiments (Appendix C). Each experiments was conducted using a single NVIDIA A30 GPU.

Table 10: Grid search reporting percentage accuracies for different message dimensions and non-linearities in an Inception-to-Inception referential game on cifar100 data.

| Non-linearity | Sigmoid | | Softmax | |
|---|---|---|---|---|
| Dimensions | Epoch 1 | Epoch 25 | Epoch 1 | Epoch 25 |
| 2 | 5.8 | 14.1 | 13.4 | 12.0 |
| 4 | 40.2 | 73.8 | 32.6 | 67.3 |
| 8 | 97.1 | 99.7 | 94.2 | 98.7 |
| 16 | 100 | 100 | 100 | 99.9 |
| 32 | 100 | 100 | 100 | 100 |
| 64 | 100 | 100 | 100 | 99.8 |

Table 11: Hyperparameters for training continuous communication channels.

| Hyperparameters | |
|---|---|
| batch size | 64 |
| optimizer | Adam |
| learning rate | 1e-4 |
| max message length | 1 |
| non linearity | sigmoid |
| vocab size | 16 / 64 |
| receiver hidden dimension | 2048 |
| image size | 384 |
| receiver cosine temperature | 0.1 |

# F  Gradient evolution during training

We verify convergence by providing evolution of the loss of individual continuous learners throughout training in Figure 8 (the same is true for discrete learners, around 4 times slower), also finding that receivers go through higher norm weight modification than senders during the initial stages of learning, as previously noted by Rita et al. (2022).

# G  Augmentations and hard distractors

We consider two training strategies commonly used in self-supervised learning: image augmentation and hard distractors. Dessì et al. (2021) showed that augmentations are crucial to develop non-degenerate protocols when training deep agents from scratch to communicate. We faithfully follow their data augmentation

Table 12: Hyperparameters for training discrete communication channels using the Gumbel Softmax.

| Hyperparameters | |
|---|---|
| batch size | 64 |
| optimizer | Adam |
| learning rate | 1e-4 |
| max message length | 1 |
| vocab size | 8192 |
| receiver hidden dimension | 2048 |
| image size | 384 |
| gumbel softmax temperature | 5 |
| gs temperature decay | 1 |
| update gs temp frequency | 1 |
| mimimum gs temperature | 1 |
| receiver cosine temperature | 0.1 |

Table 13: Hyperparameters for training discrete communication channels using REINFORCE.

| Hyperparameters | |
| --- | --- |
| batch size | 64 |
| optimizer | Adam |
| learning rate | 1e-4 |
| max message length | 1 |
| receiver hidden dimension | 2048 |
| image size | 384 |
| receiver cosine temperature | 0.1 |
| sender entropy coef | 0.5 |
| KL divergence coeff | 0 |

Figure 8: Higher gradient norms result in bigger optimizer steps. Despite being in a multi-agent setup, both agents stably converge. Receiver and sender lines on the plot are averages across all possible continuous one-to-one experiments.

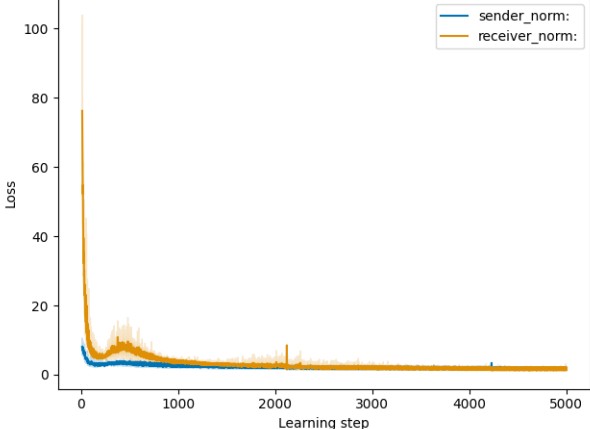

pipeline, using their same hyperparameters. We stochastically apply crop-and-resize, color perturbation, and random Gaussian blurring to every image.

Results for agents trained with data augmentations are presented in Table 14. In all experiments, including those that require generalizing out-of-domain, the agents trained with augmentations perform considerably worse than those trained without augmentations (compare to Tables 2 and 3 in the main text). We hypothesize that since, unlike Dessì et al. (2021), we are using pre-trained visual models that already abstract away from low-level features of the images, there is not further benefit in applying augmentation during the communication-training phase. At the same time, this phase only tunes a single layer, which implies that the models do not have a lot of power to adapt to the augmentations. This ends up making the task harder without bringing about better abstractions skills than those already acquired during pre-training.

Table 14: Percentage accuracy of agents evaluated on the two ImageNet validation sets (64-dimensional communication channel). Agents were trained with data augmentation applied to sender and receiver inputs. No augmentation is applied at test time

| | ImageNet1k | Single-class | OOD |
| --- | --- | --- | --- |
| **Homogeneous** | $90.2 \pm 6.7$ | $30.4 \pm 8.2$ | $61.2 \pm 13.7$ |
| **Heterogeneous** | $88.7 \pm 5.1$ | $30.2 \pm 2.3$ | $53.1 \pm 13.3$ |
| **Population** | $78.8 \pm 40.8$ | $19.3 \pm 39.5$ | $41.1 \pm 49.2$ |

Table 15: Percentage accuracy of agents evaluated on the two ImageNet validation sets (64-dimensional communication channel). Agents were trained on batches of 64 images with low cosine distance.

|               | ImageNet1k    | Single-class  | OOD            |
|---------------|---------------|---------------|----------------|
| **Homogeneous**   | $99.9 \pm 0.2$  | $47.6 \pm 2.1$  | $97.0 \pm 2.2$   |
| **Heterogeneous** | $97.1 \pm 3.8$  | $35.1 \pm 9.6$  | $68.0 \pm 18.5$  |
| **Population**    | $63.7 \pm 48.1$ | $14.2 \pm 8.5$  | $30.9 \pm 46.2$  |

Table 16: Percentage accuracy of a linear classifier trained to classify objects in the OOD dataset using communication layer representations as input. Transfer accuracy is the average percentage accuracy for the same classification task when directly transfering the trained classifier to another agent from the population.

| Vision Module  | Accuracy | Transfer Accuracy |
|----------------|----------|-------------------|
| VGG 11         | 76.7     | $87.6 \pm 2.1$     |
| Inception      | 84.6     | $89.5 \pm 3.0$     |
| ResNet 152     | 85.2     | $82.8 \pm 7.3$     |
| ViT-B/16       | 79.84    | $85.6 \pm 6$       |
| ViT-S/16 DINO  | 85.1     | $88.5 \pm 3.0$     |
| Swin           | 81.8     | $88.7 \pm 2.8$     |
| ResNet50 COCO  | 90.4     | $87.3 \pm 1.9$     |

Another interesting idea we can borrow from self-supervised contrastive learning is that of choosing *hard* distractors at training time (Robinson et al., 2021). As we do not want to rely on data annotation, we use the *cosine* distance between a target image and the other images in the training set to find near distractors (we manually checked that this simple strategy does provide intuitively similar distractors). We create batches by randomly picking an image in the Imagenet1k dataset, and adding the 63 closest images. We repeat the process throughout training. Results are found in Table 15. While agents succeed in learning communication protocols to the same degree as in the original setup, they do not outperform it. In the Single-class case, where we could expect an improvement from training on similar images, heterogeneous pairs even perform slightly worse. We thus adopt the simpler and faster random batching approach for all experiments in the main text.

## H  Exploiting the communication layer for zero-shot classifier transfer

Besides our primary goal to study communication between heterogeneous networks, a learned shared protocol is appealing from a representation learning perspective (Tieleman et al., 2018). We illustrate the point here by showing how, through this independently learned representation, we can perform seamless *zero-shot classifier transfer*, a task that is related to recent literature on "model stitching" (see references in the *Representation similarity and model stitching* paragraph of Related Work (Section 2) of the main article).

We consider a set of heterogeneous sender agents that have been population-trained to communicate on the ImageNet1k dataset. A single sender from the population is sampled and its output messages (64-dimensional vectors) are used to train a linear classifier on the OOD dataset classes. All agents reach good performance (Table 16) despite the fact that neither the communication layer nor the visual modules are further tuned to recognize the new out-of-domain classes.

The very same classifier is then transferred, without any further tuning, to another sender, and its OOD classification performance is evaluated. As shown in Table 16, the transfer senders, despite never having been paired up with this classifier during training, reach accuracy comparable to that of the classifier tested with the sender it was trained on (in several cases, the transfer senders even outperform the original senders, possibly thanks to the quality of their visual components).

Figure 9: Pairwise population communication accuracy and heterogeneous one-to-one learning time for every visual module (16-dimensional messages). The y axis denotes senders, the x axis receivers. A line separates CNNs from attention-based architectures.

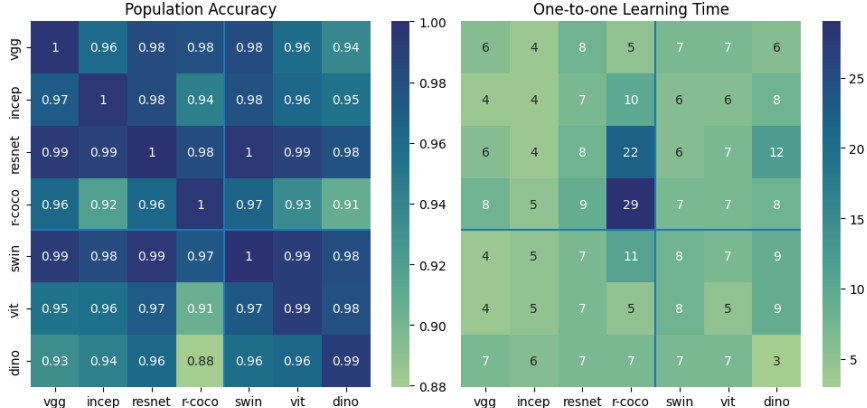

This positive result confirms that population-training leads to a single protocol shared by all senders, and it points to the usefulness of the communication layer as a compact shared representation for visual deep nets–an opportunity to be further explored in future work.

## I   Visual-module to visual-module analysis

In order to ensure that the high accuracy observed in terms of averaged population performance does not hide weaker communication for specific pairs, we checked every pair of agents' performance, and recorded them in Fig. 9. We present the accuracy matrix for the population setup when using 16-dimensional messages, since it is the one where we observe in general more variance from pair to pair, compared to the heterogeneous one, and to using 64-dimensional messages, where we observe a ceiling effect.

Communication accuracy is consistently high, even across different architecture types (CNNs: VGG11, Inception, ResNet, ResNet-COCO, left of the line; attention-based: Swin, ViT-B and ViT-S (DINO), right of the line) or different training paradigms (DINO was self-supervised, ResNet-COCO was pre-trained on the COCO dataset, while all other agents used supervised pre-training on the ILSVRC2012 ImageNet dataset). The lowest accuracies are observed when comparing ResNet-COCO (a CNN pre-trained on COCO) to either ViT-B or DINO, two attention-based architectures, suggesting a compounding effect for differences both in architecture *and* pre-training corpus.

Concerning training time, we report in Fig. 9 data for the heterogeneous one-to-one setup (population time is 28 epochs). Training times are generally low, with the two outliers involving ResNet-COCO. Note that one of these outliers is actually the homogenous ResNet-COCO-to-ResNet-COCO case, suggesting that we are observing a main effect of pre-training/communication-training corpus discrepancy, rather than issues arising when combining heterogeneous agents.

## J   Further communication errors

Further representative cases in which the 64-dimensional-message VGG 11 sender + ViT-B/16 receiver pair from the population setup failed to converge on the target are presented in Figures 10 and 11. The first sections of each figure are errors where the wrongly selected image is from the same class as the target. The transparent errors are examples where it is possible to guess the reason why the error arose, while the opaque errors are unclear as such, and would require some more investigation.

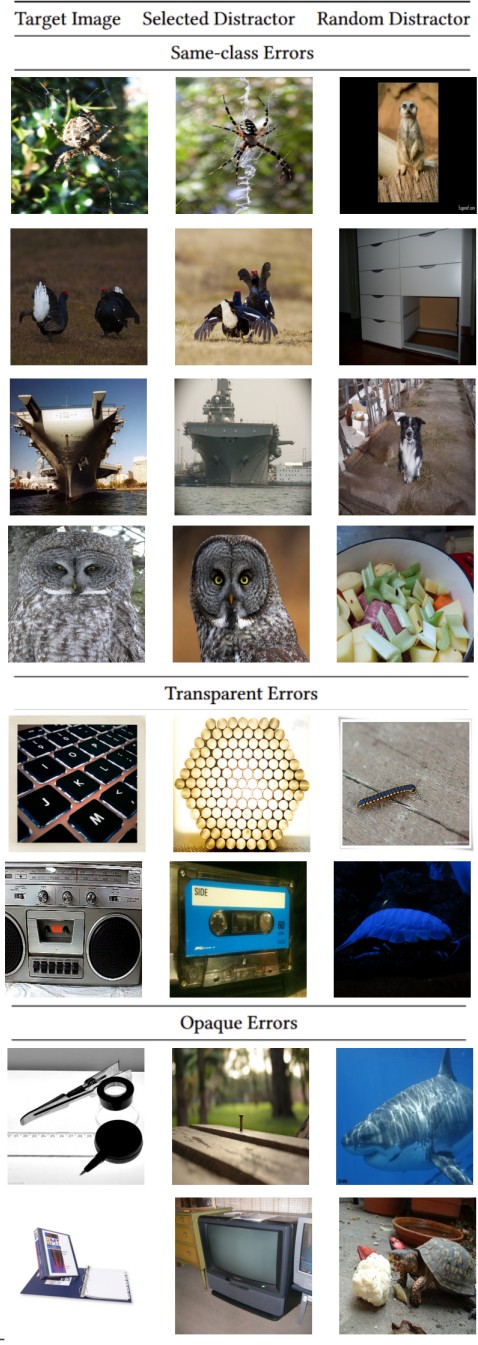

Figure 10: Examples from the ImageNet1k-based test set of mistakes made by a VGG 11 sender + ViT-B/16 receiver pair trained in the population setup. We also show an additional random distractor from the batch for context.

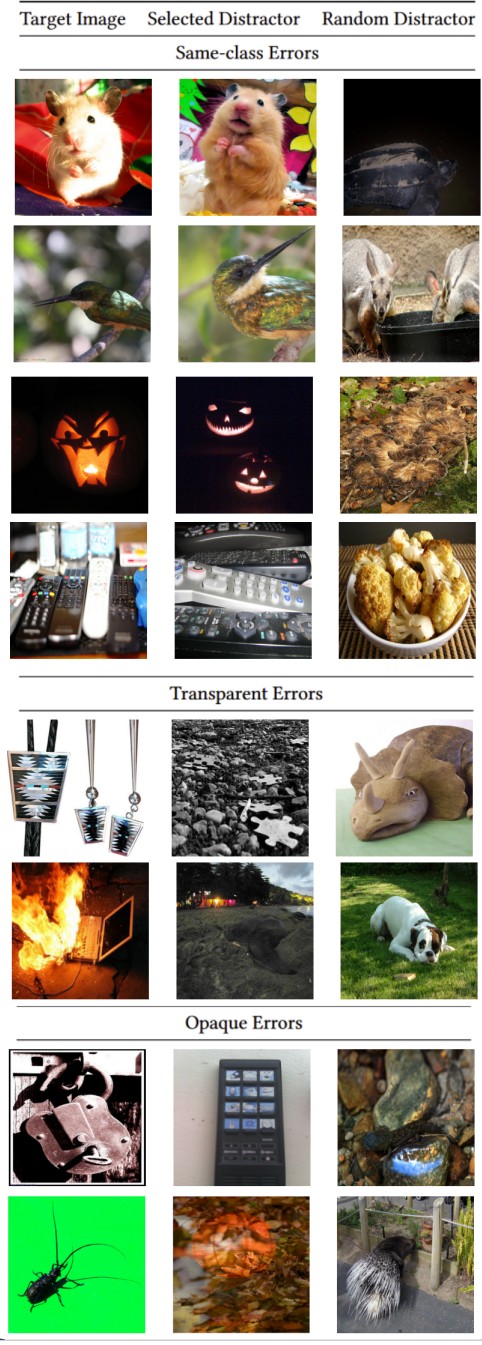

Figure 11: Examples from the ImageNet1k-based test set of mistakes made by a VGG 11 sender + ViT-B/16 receiver pair trained in the population setup. We also show an additional random distractor from the batch for context.

Figure 12: Explained variance for a PCA across messages (left) and correlation of the different continuous dimensions (right) from a population setup. Messages were generated by senders exposed to the ImageNet1k test set.

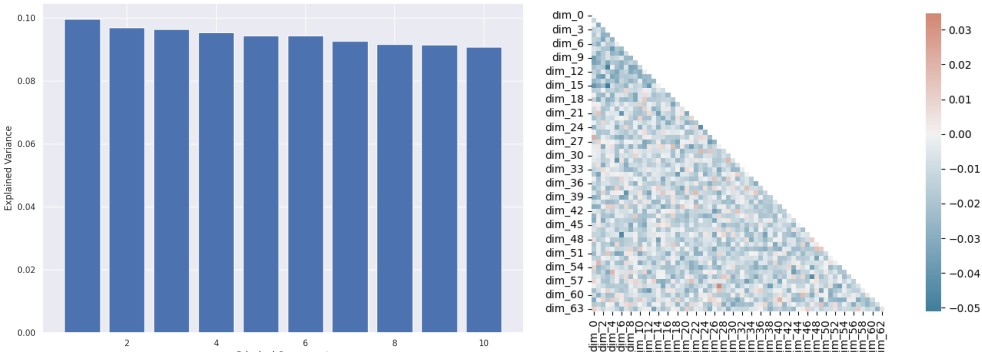

## K   Discretization and other channel perturbations

In the spirit of Carmeli et al. (2022), we wondered whether we could use an easy-to-optimize continuous channel at communication training time, but then discretize the messages at inference time, thus getting the "best of both worlds".

We communication-trained the agents with dense continuous messages, but at evaluation time we discretized their messages by rounding the value of each continuous dimension in the message to either 0 or 1. On the ImageNet1k test set, performance fell to near chance levels for all investigated thresholds, in both population and one-to-one settings. Performance loss was also drastic when transforming both the 16 and 64-dimensional continuous message into a one-hot vector, using the maximum value as the "hot" value.

The continuous messages thus seem to be genuinely dense. Indeed, further analysis showed that introducing Gaussian noise (e.g., $\mu = 0$, $\sigma = 0.05$) in the messages during evaluation impacted performance negatively (22% accuracy loss). Very small changes in the message representation significantly changed the image the receiver would choose.

That information is not sparsely encoded in the continuous vector dimensions, or in some latent combinations of them, is confirmed by a PCA analysis over 64-dimensional message space. The PCA eigenvalues are shown in Fig. 12. All PCA dimensions explain similar amounts of variance, with no dimension holding significantly more information. The figure also shows that there is no significant correlation between the original continuous vector dimensions, suggesting that they all carry independent information.

