# OpenReview forum: "Referential communication in heterogeneous communities of pre-trained visual deep networks"
_TMLR — Rejected by TMLR_

### Review · Reviewer_qz7E · 2024-04-07

**Summary Of Contributions:**

The paper investigates whether it is possible to learn a communication protocol for a community of pretrained visual models so that they can refer to a specific image within a set of images. In contrast to related works that consider pretrained models of the same architecture, this paper considers pretrained models of different architectures, sizes, and training methods. The paper proposes a simple method that achieves >95% accuracy on in-distribution images, while leaving room for improvement in out-of-distribution generalization.

**Audience:**

Yes

**Claims And Evidence:**

Yes

**Requested Changes:**

Please see weaknesses, especially the first point.

**Strengths And Weaknesses:**

Strengths:
- The paper is well written and easy to follow.
- The topic of communication among heterogeneous pretrained models is interesting and well motivated.
- The experiments present quite extensive analysis, including community types, generalization to difficult distractors and unseen images, and the possibility of incrementally adding agents to a trained community.

Weaknesses:
- Some observations made in the experiments can be strongly influenced by the pretrained model. The paper does not clearly separate the effects of the pretrained model and the added communication module. For example, regarding the limited generalization ability, is it because the pretrained model itself cannot generalize well, or because the low-dimensional protocol loses critical information? Regarding the “Gaussian blobs” test, one can also argue that it is because the pretrained model already captures high-level semantics and discards pixel-level details.
- Some clarity issues:
    - Fig. 1 seems to be in low resolution, and becomes blurry when zoomed in.
    - It would be better to have an equation showing the training loss, and some pseudocode summarizing the training procedure.

---

> ### Author Response · Authors · 2024-05-21
> **Response to Reviewer qz7E**
>
> Thanks for your feedback, that made us reflect more thoroughly about the dichotomy between pretraining and communication training.
>
> *Some observations made in the experiments can be strongly influenced by the pretrained model. The paper does not clearly separate the effects of the pretrained model and the added communication module. For example, regarding the limited generalization ability, is it because the pretrained model itself cannot generalize well, or because the low-dimensional protocol loses critical information? Regarding the “Gaussian blobs” test, one can also argue that it is because the pretrained model already captures high-level semantics and discards pixel-level details.*
>
> We agree that we did not address the role of pre-training clearly enough. In part, this is because we are interested in the scenario in which we need to let existing pre-trained networks communicate, so it is a basic assumption that we are dealing with pre-trained networks.
>
> Earlier work (Dessi et al NeurIPS 2021) found that  when networks are trained from scratch for communication only, they do not pass the Gaussian-blob test, unless augmentations are used. This suggests that, indeed, pre-training plays a crucial role in capturing high-level semantics. Note however that zero-shot performance of a pre-trained network on our OOD and same-class tests would be at chance level, showing that communication-training is clearly beneficial.
>
> To address more formally the extra-benefit of communication training with respect to basic pre-training, we added a new experiment in Appendix B. We played the OOD discrimination game using as messages the classes obtained by unsupervised clustering of last-layer representations of the OOD data-set itself. We found that performance, while not at random level (in the only meaningful case, namely the homogenous one), lies much below that of communication-trained agents, confirming that the latter adds crucial discrimination capabilities that are missing from pre-trained representations
>
> We have moreover added a brief remark to section 4.1 to clarify that we do indeed believe that pre-training is playing an important role in extracting more abstract representations that facilitate the heterogeneous communication setup. At the same time, as we discuss at the end of the same section, there is no strong correlation between similarity of underlying pre-trained representations and communication game performance, once more suggesting that communication-based training is adding non-trivial information.
>
> *Fig. 1 seems to be in low resolution, and becomes blurry when zoomed in. (done)*
>
> We fixed it.
>
> *It would be better to have an equation showing the training loss, and some pseudocode summarizing the training procedure.*
>
> We added pseudo-code and training loss equation in Appendix A.

---

### Review · Reviewer_FX82 · 2024-04-14

**Summary Of Contributions:**

The paper studies referential communication between different pre-trained visual models. Given a set of images and two (frozen) vision models, lightweight adaptors are trained on top of the models to extract compact representations (“codes”) and the target is for the second model (“receiver”) to select the correct image based on these codes. The paper studies this setting in some detail and in particular, explores “population training” where many models are trained to communicate simultaneously.

**Audience:**

Yes

**Broader Impact Concerns:**

no specific concerns

**Claims And Evidence:**

No

**Requested Changes:**

Address the cons
- clarify the training setup and very desirably fix/improve it - otherwise the presented results may be incomplete/misleading (Con 1)
- perform experiments with larger dimensionality of the code - otherwise the presented results may be incomplete/misleading (Con 2)
- address the presentation issues (Con 4)
- (less crucial) consider adding a CLIP model to the comparisons

**Strengths And Weaknesses:**

Pros:
1. An interesting and fairly original setting: communication research is not at all new, but this is an interesting take on it.
2. Fairly thorough experiments, including 1-to-1 and population training and experiments on introducing new models to an existing protocol
3. Reasonable analysis of generalization to other classes, perturbations

Cons:
1. Some doubts/questions about the training protocol:
1a. “leading to 781 batches of 64 images” - this sounds like batches are fixed and not resampled during training, is that so? It would be quite strange and unconventional.
1b. It appears that no data augmentation has been applied at training time - that's also quite unconventional, why is it? It’s very standard practice and I don’t see how it would hurt, should only make communication more robust. It would be interesting to compare applying the same transformation to all images in a batch vs different transformations.
1c. It appears that training batches are sampled randomly? If so, given that ImageNet has many classes, a very simple strategy for the model would be to just communicate the class - as evidenced by the same-class experiment, this happens to a large degree. Why not sample batches at training time smarter, such that same-class examples are often seen during training and the model has to learn to communicate the differences?
2. Somewhat related, the selection of the code dimensionality to be 16 seems quite unjustified. Tuning the dimensionality of the code on CIFAR-100 seems strange given that all other experiments are on ImageNet - since CIFAR-100 is a simpler dataset, one might very well end up setting an overly low dimensionality? The interesting question to be investigated here is if there are downsides to low dimensionality, e.g. in terms of adaptation to new classes and new networks being added, or becoming very “low-level” (which, to some degree, is not necessarily such a bad thing, in my opinion). If not, why not just set it higher?
3. I'm surprised to see that there are no CLIP-style models in the list of so compared models, why is that? Adding those could be especially interesting since it might open doors to mapping the communication protocol to natural language.
4. Presentation issues.
4a. Wrong first author for the citation of the ViT paper (and it’s twice in the bibliography).
4b. Figure 1 - please increase resolution, at the moment looks unprofessional and somewhat difficult to read.
4c. Please add more qualitative examples. And it appears images are oversaturated in Figure 9? Please fix.
4d. The paper says color perturbations “don’t have a very strong effect on accuracy” - I wouldn’t agree here, in Figure 4 the accuracy seems to go from maybe 98-99% to maybe 93%, which is ~5x increase in error rate. And it’s even more for other perturbations. This is a very strong effect.
4e. “the population has truly developed a single protocol” is an overstatement: for sure the messages are much more similar than for one-to-one training, but still especially in some cases very different, >0.3 distance. Please tone down.
4f. The setting seems related to image retrieval, please discuss it in the paper.

---

> ### Author Response · Authors · 2024-05-21
> **Response to Reviewer FX82**
>
> Thanks for suggesting some very interesting experiments!
>
> *Clarify the training setup and very desirably fix/improve it - otherwise the presented results may be incomplete/misleading (Con 1)*
>
> We expressed ourselves poorly. Batches are indeed not fixed, but resampled during training. We removed the confusing text.
>
> We ran experiments with data augmentation (Appendix G), finding that it leads to a *decrease* in performance across experiments. We hypothesize that, as the pre-training pipeline already makes the networks capable of extracting more abstract features from the networks, they are not useful in the communication-training phase. Furthermore, since the benefits of augmentations are already acquired during pre-training, in this second phase they only make the task harder (during this phase, only a single layer is trained, not giving the models a lot of power to adapt to the transformations). It is possible that, by using different hyperparameters for the augmentations, we might ultimately find a setup where they might help, but it seems clear from the current experiments that they are not as beneficial as in basic self-supervised pre-training.
>
> Concerning training with hard distractors, we prefer not to use ImageNet classes to pick the latter, as we want to keep our setup fully self-supervised. We ran instead experiments where hard distractors for training are picked based on their cosine similarity to the targets in image space (Appendix G).  It turns out that this does not improve performance, not even in the same-class experiment, where one could have expected it would.
>
> *Perform experiments with larger dimensionality of the code - otherwise the presented results may be incomplete/misleading (Con 2)*
>
> We originally used the CIFAR100 validation set to pick the dimensionality hyperparameter. As we now explain in Appendix D, we chose this validation set since we are using the one from ILSVRC2012 for testing. We found that all dimensionalities >=16 reach 100% performance. We thus reasoned that, all else being equal, it made sense to pick the smallest dimensionality (16), and we used that for the original experiments.
>
> However, the reviewer has a good point in stating we might be observing a ceiling effect due to the simplicity of CIFAR100, which might be masking differences between shorter and longer messages. We thus decided to run all the main experiments in the paper also with dimension 64 (the largest dimension we explored). We found that agents trained with 64-dimensional message vectors generally outperform those trained with 16-dimensional messages, even in the generalization setups, where we would have expected the smaller dimensionality to act like a beneficial bottleneck.
>
> *Address the presentation issues (Con 4)*
>
> - We fixed the issue with the ViT reference.
>
> - We carefully rephrased the claims we make in the discussion of how perturbations affect accuracy, no longer focusing on absolute accuracy, but on the comparison between Gaussian blur and the other conditions.
>
> - We toned down the claim about population-trained agents having “truly” developed a single protocol.
>
> - We now acknowledge the relationship between our task and content-based information retrieval in the Related Work section.
>
> - We have redrawn Figure 1 with increased resolution and fixed oversaturation in Figure 9.
>
> - We extended the qualitative examples.
>
> *(less crucial) Consider adding a CLIP model to the comparisons*
>
> In a parallel line of research, we are exploring the idea of using a referential game to fine-tune CLIP-like-based caption generation and retrieval architectures. However, we deliberately excluded CLIP-like encoders from the current study, because we feel that the fact that such models have been trained with natural language adds a further important axis of variation that deserves its own study. We added it as a good direction for future work in the conclusion.

---

### Review · Reviewer_hYQh · 2024-04-28

**Summary Of Contributions:**

This paper contributes by systematically exploring how diverse pre-trained visual networks autonomously develop shared communication protocols despite architectural differences, demonstrating their adaptability to unseen object categories and ease of protocol acquisition, while also analyzing the emergent protocol's properties, thereby advancing understanding of referential communication among heterogeneous networks.

**Audience:**

Yes

**Claims And Evidence:**

Yes

**Requested Changes:**

1. It would be beneficial for the paper to provide more details about the experimental setup, particularly regarding the selection process of negative candidates.
2. It could offer valuable insights into the robustness and adaptability of communication protocols across varying training data by exploring how agents trained on diverse datasets perform in the referential communication task.
3. Additional insights into the communication task would be appreciated.

**Strengths And Weaknesses:**

strengths:
The topic of referential communication in heterogeneous communities of pre-trained visual deep networks is indeed intriguing. Exploring how these networks can develop shared communication protocols despite their varied architectures and training regimes offers valuable insights into the potential for autonomous agents to interact effectively.

Weaknesses:
The high accuracy achieved in the referential communication game task is quite surprising and noteworthy. It raises some questions:
1) how this work process of selecting candidate images, particularly negative images, and how they contribute to the referential communication task?
2) Any further insights into the strategies and processes involved in achieving this level of accuracy?
3) The paper mentions that nearly all agents rely on vision modules pre-trained on the ILSVRC2012 training set. It seems not a common assumption in the real world. It raises questions about the generalizability of the findings to agents trained on different data distributions.

---

> ### Author Response · Authors · 2024-05-21
> **Response to Reviewer hYQh**
>
> Thanks for your constructive feedback!
>
> *It would be beneficial for the paper to provide more details about the experimental setup, particularly regarding the selection process of negative candidates.*
>
> As we now clarify in the paper, distractors in the main experiments are selected randomly. We ran a new experiment in which we train using hard distractors that have high input-image cosine similarity to the target. Results suggest that this strategy does not improve overall performance with respect to the less laborious and faster approach in which we use random image batches. The experiments are reported in Appendix G.
>
> *It could offer valuable insights into the robustness and adaptability of communication protocols across varying training data by exploring how agents trained on diverse datasets perform in the referential communication task.*
>
> We would love to experiment with networks pre-trained on *very* different data-sets, but we have not found trustable publicly available models of this sort (and we do not have enough resources to train them in-house).  In the paper, we do report results with a ResNet trained on COCO (whereas all other models were trained on ImageNet). We find that, in general, this network is harder to train to play the referential game (even in the homogenous case, in which we are using it both as sender and as receiver). However, once this difficulty (which might be related to the smaller and simpler nature of the COCO pre-training data) is taken into account, it does not appear that the COCO-trained network has any special problem interacting with ImageNet-trained networks, tentatively suggesting that communication is robust to pre-training dataset diversity. We discuss all this in section 4.1 and Appendix I.
>
> *Additional insights into the communication task would be appreciated.*
>
> We added more qualitative examples to Section 4.4 and in Appendix J.

---

### Decision · Action_Editor_gKWG · 2024-06-24

**Recommendation:** Reject

**Comment:**

The paper explores the task of referential communication in a community of heterogeneous pre-trained visual networks, aiming to develop a shared communication protocol despite varying architectures and training regimes. The study shows promising results in enabling these networks to refer to target objects among a set of candidates, generalize to previously unseen object categories, and integrate new networks into an existing community protocol. The research provides qualitative and quantitative evidence suggesting that the emergent protocol captures high-level semantic features of objects.

This paper investigates a very novel and practically valuable topic. Proposing and exploring this new topic is one of the core contributions of this paper. The authors have also provided many seemingly interesting experimental setups, and some of the current experimental results suggest that this is a worthwhile direction of research. However, the paper has serious shortcomings in the sufficiency and completeness of the experiments, significantly differing from its claimed systematic experimentation. Regarding several innovative experimental setups, the descriptions in the paper are not clear enough, leading multiple reviewers to question the solidity of its conclusions. This raises significant concerns about the generalization and adaptability of the findings. In further work, the authors should pay special attention to the reproducibility of the experiments, as well as the presentation and clarity. They should also compare as many types of models as possible, including self-supervised learning models, CLIP-based models, and reinforcement learning models.

The paper has a significant potential and contributes to a novel and important area of research. However, the issues highlighted by the reviewers are critical and must be thoroughly addressed to strengthen the validity and clarity of the findings. I invite the authors to revise the manuscript accordingly, addressing each of the weaknesses and requested changes in detail. Upon resubmission, the revised paper will be re-evaluated to ensure that the necessary improvements have been made.

**Audience:**

This paper may be insightful for readers in the following three areas. Firstly, it introduces a new research direction, which significantly differs from existing topics. Those interested in novel research directions may benefit from it. Secondly, researchers focusing on Heterogeneous Systems may find some useful ideas in this paper. Additionally, those interested in practical applications like autonomous driving might also find this paper appealing.

**Claims And Evidence:**

Claim 1:
This paper claims “systematically explore the task of referential communication in a community of heterogeneous state-of-the-art pre-trained visual networks”. Especially, it intends to study practical scenarios such as “two self-driving cars from different makers, that need to coordinate about objects in their environment” and “One model powered by a ResNet trained on the ImageNet visual recognition challenge, whereas the other use a Vision Transformer trained with a self-supervised algorithm.”

Lack of Evidence 1:
This paper indeed studies some different architectures. To be specific, it selects four convolutional networks and transformers. However, the model chosen in this paper has significant limitations. The model types are constrained. It also includes different size variants of the same model type (ResNet and ViT). Many state-of-the-art models have not been compared, including DenseNet, CaiT, Levit, and ResNext. Additionally, in terms of data sources, most models only used standard ImageNet training, with only one instance used the COCO dataset. This point was also noted by Reviewer hYQh. The authors did not compare self-supervised methods as stated. It should also be noted that the authors mentioned how cars with different autonomous driving algorithms interact, mainly involving the models related to control and reinforcement learning. However, the content of this paper is limited to perception models. This is one of the discrepancies between the actual content of the paper and its stated focus. Furthermore, Reviewer FX82 pointed out that the paper did not explore CLIP-based models. The authors' response to these issues mainly describes a lack of computational resources, but this is not convincing since these models usually do not require training. The authors also stated that they are exploring related experiments but could not provide new results. The experimental limitations significantly restrict the feasibility of this paper in terms of generalizability and adaptability.

Claim 2:
This paper explores the potential interaction scenarios of different models in practical application settings. Specifically, it provides three clear conclusions: “1) we show that it is indeed possible for sets of heterogeneous pre-trained networks to successfully converge on a referent through an induced communication protocol. 2) We study referential generalization, showing that the developed protocol is sufficiently flexible that the networks can, to some extent, use it to refer to objects that were not seen during the training phase, as well as to tell apart distinct instances of the same object. 3) We consider instead agent generalization. We show that, given a community of networks that have been trained to communicate, it is faster for a new pre-trained network with a different architecture, training objective or training corpus to acquire the existing community protocol, than for the enlarged community to develop a new protocol from scratch.”

Lack of Evidence 2:
The above description is rather abstract and needs to be understood in conjunction with specific experimental setups and results. As all the reviewers pointed out, the paper lacks clarity in experimental details, with many details not being presented. Reviewer hYQH noted that the high accuracy in the referential communication game task is surprising. Reviewer qz7E pointed out that some conclusions might stem from the pre-trained model itself rather than the added communication module. Therefore, the three main arguments presented in the paper lack a sufficiently comprehensive experimental support.

**Resubmission Of Major Revision:**

The authors may consider submitting a major revision at a later time.